# Nano-and Microparticles of Carbon as a Tool for Determining the Uniformity of a Diffuse Discharge Exposure

**Mikhail Lomaev** [ID], **Victor Tarasenko \*** [ID], **Mikhail Shulepov, Dmitry Beloplotov** [ID] **and Dmitry Sorokin** [ID]

Institute of High-Current Electronics SB RAS, Tomsk 634055, Russia
\* Correspondence: vft@loi.hcei.tsc.ru; Tel.: +7-903-953-9631

**Abstract:** At present, a diffuse discharge plasma of air and other gases at atmospheric pressure is widely used for the surface treatment of various materials. However, in many papers it is stated that erosion damages occur on flat anodes (targets) as a result of the discharge plasma action. The shape of these damages depends on the discharge mode. In this study, the exposure uniformity was investigated by using nano- and micro-sized carbon particles deposited on a flat copper anode (a carbon layer). The diffuse discharge was formed in a 'point-plane' gap with a non-uniform electric field strength distribution by applying voltage pulses with an amplitude of 18 kV. It has been established that at a gap width of 8–10 mm, an imprint of the discharge plasma on the carbon layer deposited on a copper anode has no traces of local erosion. In order for erosion to occur on the surface of the anode in the form of uniformly distributed microcraters, it is necessary to increase the current density at the anode, for example, by decreasing the gap width. When decreasing the gap width to 6 mm and less, spark channels occur. They damage both the carbon layer and the copper anode in its central part. It has been shown that there are three characteristic zones: a color-changing peripheral part of the carbon layer, a decarbonized central part of the anode, and an annular zone located between the central and peripheral parts and containing individual microcraters.

**Keywords:** pulsed diffuse discharge; flat anode; carbon micro- and nanoparticles; impact heterogeneity; impact uniformity; spark channels

## 1. Introduction

Technologies based on a 'cold' non-equilibrium low-temperature plasma, including those incorporating micro-and nanoparticles, are widely used and are constantly being improved [1–4]. Particles of various kinds and sizes are involved in plasma treatment processes [5–7]. So, carbon particles are used to solve a number of problems, in particular, in coating technologies, e.g., [8]. As a result, durable coatings are formed on an anode surface (a target; a sample), providing better performance. Such a coating, for example, can significantly increase the service life of tools or machine components.

Carbon nanoparticles can also affect the electron emission from a cathode. It was shown in [9,10] that the formation of a film composed of carbon nanoparticles on the cathode surface leads to a decrease in the threshold electric field strength necessary for the field emission onset. In addition, such films are used to control the exposure uniformity on the anode surface at pulsed and pulse-periodic discharges in a non-uniform electric field. In [11] and other papers, a carbon coating (soot) was used to increase the sensitivity of a detection method for microchannels developing at the initial stage of the discharge.

In [12–15] it was shown that a great number of microdamages are formed on the surface of anodes made of various metals. At the same time, with a decrease in the energy input into the discharge plasma, these damages were well visualized using a carbon film deposited on the anode surface during the combustion of gasoline of a lighter. The characteristics of a voltage pulse, gap width, and anode material determine the size of the damaged area. For metal anodes, the diameter of the damaged area varies in the range of

1–20 µm. In [11] using multi-frame laser interferometry, it was established that erosion of an electrode is associated with the appearance of microchannels in the diffuse plasma bulk. It was proposed there that the regions occupied by microchannels are characterized by a higher temperature and a lower particle concentration, which makes it easier to reach the threshold reduced electric field strength for the generation of runaway electrons and, as a consequence, bremsstrahlung X-rays.

Formation of microchannels in a pulsed discharge plasma was also observed in [16–19], where jets with high concentration of particles occurred after appearance of spark channels near the electrodes. These studies were carried out in air-filled mm-length 'point-plane' gaps with a non-uniform electric field strength distribution and at both polarities of the pointed electrode. A confirmed fact that at first bright spots appear on an electrode with a small radius of curvature, see, e.g., [20]. Then, after a short time period, bright spots arise on the flat electrode (cathode). These spots are the starting points of spark leaders, which, after gap bridging, transform into spark channels. Due to the high temporal resolution of a method used in [16–19], it was found that at small interelectrode distances and strong electric field near the pointed electrode the breakdown of the gap occurs as a result of the rapid (less than 1 ns) formation of micro-sized cathode and anode spots. It has been shown that these spots are highly ionized plasma (electron concentration is $n_e \sim 10^{19}$–$10^{20}$ cm$^{-3}$). This plasma gives rise to spark channels of small diameter. However, the authors of [16–19] did not deal with primary streamers ($n_e \sim 10^{14}$ cm$^{-3}$) and a diffuse discharge phase which take place before the spark transition. In addition, they did not mention about runaway electrons and bremsstrahlung X-rays that promote to the formation of the diffuse discharge in atmospheric pressure air and other gases/gas mixtures. It is known that pulsed diffuse discharges initiated by runaway electrons and bremsstrahlung X-rays caused by these electrons [11] are used for the surface modification of different materials, see, e.g., [21].

Studies presented in [22,23] should also be mentioned, where the breakdown stage of a discharge in a non-uniform electric field was studied. In [22], a gap formed by two pointed electrodes and filled with nitrogen or a mixture of nitrogen and water vapors was broken down by high-voltage pulses. It was established with a high-speed ICCD camera that the breakdown starts from the formation of a streamer near the high-voltage pointed electrode of positive polarity. This streamer provides the formation of a diffuse discharge, which subsequently passes into the spark phase. In [23], to study the development dynamics of the discharge in air and nitrogen, an ultrafast streak camera was used. It was supposed that "spark discharge emission emerges in a streamer-free spontaneous manner and decays in two distinct phases". The first phase lasts about 2 ns and plasma radiation within it emitted essentially due to the spectral transitions of the second positive and first negative systems of a nitrogen molecule $N_2$ and molecular ion $N_2^+$, respectively (SPS and FNS systems). Radiation of NO* molecules and the first positive system (FPS) of $N_2$ from the discharge plasma are typical for the second one. These facts indicate that the first stage is the diffuse discharge [20]. Both in [22,23] the characteristics of runaway electrons and bremsstrahlung X-rays were not registered.

From the analysis of known papers, it follows that low-temperature plasma of diffuse discharges in different gases at atmospheric pressure, including ambient air, is extremely promising for practical use. Nevertheless, the formation of microchannels and damages on a target (anode) can limit the number of applications of diffuse discharges in a non-uniform electric field. From the other hand, as already mentioned, such discharges in air and other gases at atmospheric pressure were formerly used for the cleaning, oxidation, and hardening of the surface of solids (anodes) [21]. It should be noted that in some of cases, no damages were observed on plasma treated anodes made of aluminum, copper, and other metals [21,24]. The most common way to implement the diffuse discharge in a dense gaseous medium (air or other gases) without the use of external sources of ionizing radiation is to feed the 'point(cathode)-plane(anode)' gap placed in this medium with high-voltage short-duration pulses. There are several factors affecting the anode in this case. First, a dense plasma enriched with reactive species. Second, ultraviolet (UV) and vacuum

ultraviolet (VUV) radiation emitted by exciting species. Third, shockwaves arising due to the fast energy deposition. In addition, a (sub)nanosecond rise time of voltage pulses contributes to the generation of runaway electrons and, consequently, bremsstrahlung X-rays ensuring a diffuse form of the discharge, as well as affecting the anode surface. Nevertheless, in a wide range of conditions, even at the nanosecond rise time of a voltage pulse, the diffuse stage is limited by the discharge constriction [16–20], which has a local destructive effect on the anode surface [11–15]. Moreover, damage to the surface of electrodes is observed when bright spots appear on them. Therefore, it is important to carry out additional studies aimed at studying the effect of the diffuse discharge formed in a non-uniform electric field on the surface.

In this regard, the goal is to study the influence of exposure modes of the pulsed diffuse discharge plasma formed in a non-uniform electric field on the surface of a flat anode. Different gap widths made it possible to change the exposure mode, as well as the specific energy input into the discharge plasma. To increase the sensitivity of the method for detecting erosion of a flat electrode, its surface was coated with a layer of carbon micro- and nanoparticles (soot). The preliminary results of our research are presented in [25].

## 2. Experimental Setup and Methods

The experiments were carried out on an experimental setup sketched in Figure 1. It consisted of an NPG-18/3500N high-voltage (HV) generator (1) [26], a 3-m-length 75-Ω HV cable (2), a low-inductance shunt of reverse current (3), a discharge chamber (4) with an electrode assembly formed by a HV cathode with a small radius of curvature (5) and a flat grounded anode (6), a Sony A100 digital camera (9), a TDS MDO3102 digital oscilloscope with a bandwidth of 1 GHz and a sampling rate of 5 GSa/s (10), and a desktop computer (11). The characteristics of a voltage pulse applied to the gap are the following: the rise time is ≈4 ns, a duration (full width at a half-maximum; FWHM) is ≈8 ns, and an amplitude is ≈18 kV. The cathode made of a 5-mm-diameter stainless steel rod had a pointed end with an apex angle of ≈50° and a the radius of curvature of ≈0.2 mm. The grounded anode plate was made of a 25 mm diameter and 1 mm width copper disk. The gap width $d$ was varied from 2 to 10 mm.

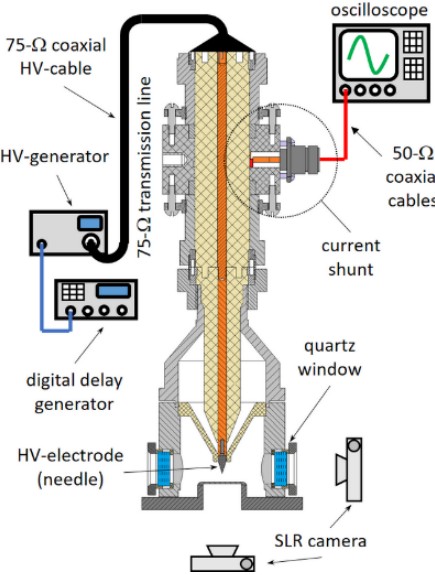

**Figure 1.** Sketch of the experimental setup.

To increase the sensitivity of the imprint method (discharge plasma exposure traces), the anode surface facing the cathode was covered with carbon micro- and nanoparticles (soot). The flame of a gasoline lighter was used to deposit soot. Figure 2a demonstrates a glass plate covered by individual carbon particles and clusters consisted of such particles.

The image was captured with a POLAR-1 microscope (Micromed). To determine the size of individual carbon particles, a photo of the transparent glass plate was captured at the beginning of the soot deposition process, i.e., when the first particles and clusters, which had different sizes, fell on the plate. At the same time, part of the plate was not covered with carbon particles and remained transparent. The minimal size of carbon particles was several hundreds of nanometers. Such particles formed a soot layer. Experiments on the discharge plasma exposure were carried out with a film (carbon layer) having a thickness of ~20 μm. With such a film thickness, it was opaque to visible radiation. The thickness of the carbon layer was determined from the that of the layer around the crater formed as a result of damage during the discharge with bright sports on anode.

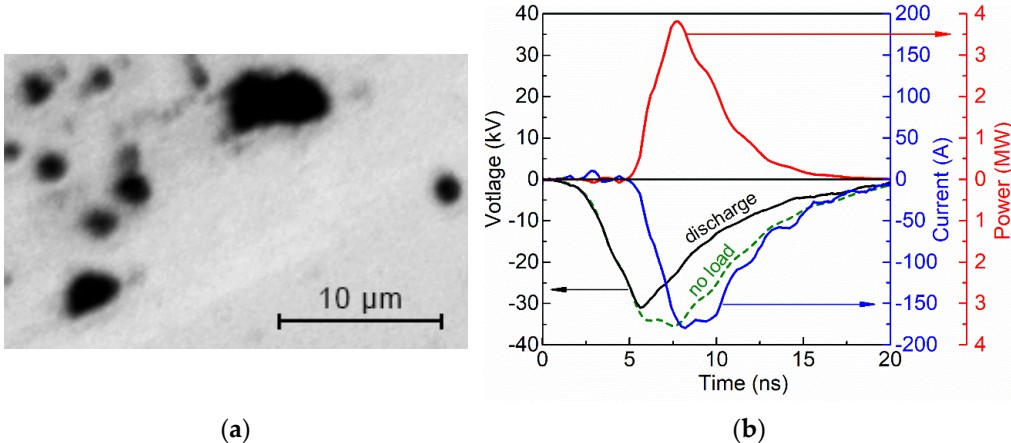

| (a) | (b) |

**Figure 2.** Individual carbon particles and carbon clusters (**a**). Waveforms of the voltage during discharge and in a no-load mode, as well as waveforms of discharge current and discharge power (**b**).

Discharge was formed in the air-filled gap at pressure $p$ = 1 atm. The air humidity was 23–35%. The generator operated in the single-pulse mode. Figure 2b demonstrates waveforms of the voltage across the gap $U$ (black curve), the idle voltage $U_0$ (dotted curve) and discharge current $I$ (blue curve) recorded with the reverse current in the circuit. The resistance of the current shunt was 0.014 Ω. To reconstruct $U$ and $I$, the waveform of an incident voltage wave (not shown) recording with a capacitive voltage divider mounted in 75-Ω transmission line and signals from the reverse current shunt were used. In addition, based on $U$ and $I$ the time behavior of the discharge power $P$ was calculated (Figure 2b, red curve). It follows that the energy deposited to the discharge plasma within 15 ns was ≈ 17 mJ for $d$ = 6 mm. This value is ~70% from the energy of that of the 18-kV-amplitude voltage pulse from the generator.

Integral images of the discharge plasma glow were captured with the SLR camera through side quartz windows of the discharge chamber. The state of the anode surface after a single discharge plasma exposure at different $d$ was inspected with an MBC-10 microscope (LOMO).

In these experiments, compared to [25], a special anode unit with a transparent conducting surface was used. This made it possible to capture the discharge plasma glow through it in various planes located perpendicular to the discharge axis, as well as the near-cathode and the discharge glow with particles in the plane of its tip.

Additionally, a four-channel ICCD camera was used for recording the plasma glow dynamics at the stages of discharge formation and combustion. The camera operated in two modes. The first is standard mode (sequential recording of individual frames with a gate of 3 ns). The second is the high-temporal-resolution mode. The latter was achieved due to the joint action of the synchronization accuracy of the HV generator and the ICCD camera, as well as setting the minimum possible delay between frames (more detailed in [27]). The streamer dynamics at the breakdown stage (several nanoseconds) was established owing the second mode (Figure 3).

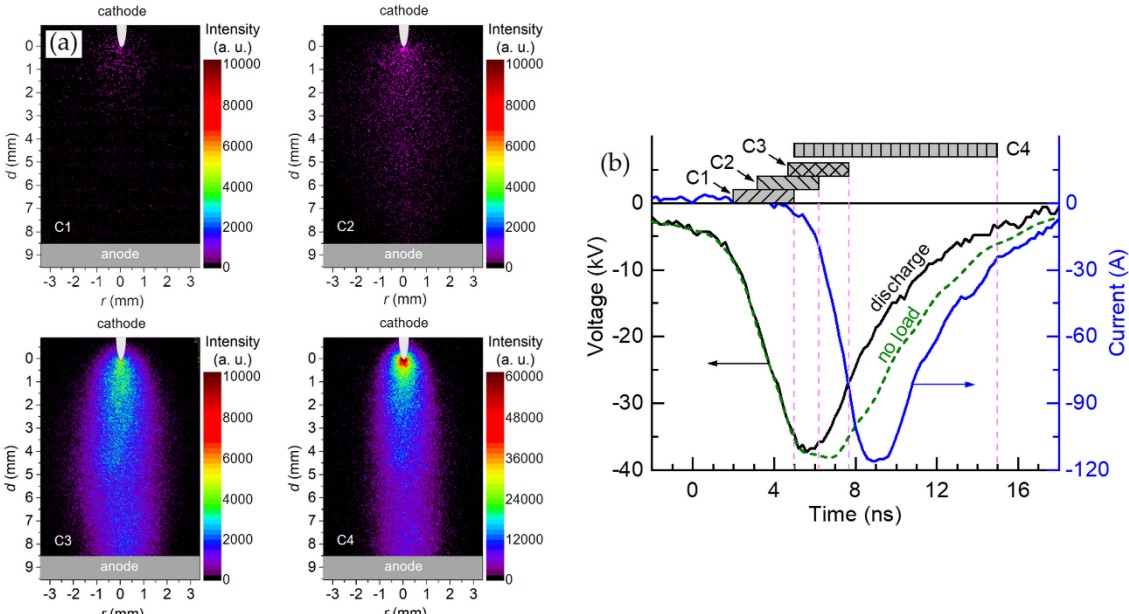

**Figure 3.** Time resolved images of the discharge plasma glow in atmospheric pressure air captured with the ICCD-camera (**a**) and waveforms of the discharge current (blue curve) and voltage across the gap during discharge and in a no-load mode (**b**). HV generator NPG-18/3500N, $d$ = 8.5 mm. C1, C2, C3 and C4 show the duration of each frame and their time position relatively the voltage and current waveforms.

## 3. Results

### 3.1. Diffuse Discharge as a Result of the Breakdown of Atmospheric-Pressure Air in Non-Uniform Electric Field by Nanosecond Voltage Pulses

A great number of scientific papers are devoted to diffuse and spark discharges developing in accordance with the streamer breakdown mechanism [27–32]. The results of studies of the breakdown of air at atmospheric pressure in a non-uniform electric field depend significantly on the design of electrodes, the characteristics of a voltage pulse, and the pulse repetition rate. Further, we deal with discharges implemented without additional preionization sources in gaps formed by an HV cathode with a small radius of curvature and a flat grounded anode. This polarity ensures the most diffuse form of a discharge (homogeneous morphology; see, e.g., [32]).

When the distribution of the electric field strength is non-uniform, the discharge starts from a pointed cathode. This is due to the field emission whose threshold decreases because of microinhomogeneities on the cathode surface [33]. An additional contribution to this decrease is made by micro- and nanoparticles, in particular carbon ones, deposited on the cathode surface [9,10]. Then electron avalanches are formed, whose heads overlap near the cathode tip due to the electric field amplification there. Reaching the critical number of electrons in an avalanche [34] leads to the appearance of a streamer. Under conditions of a non-uniform electric field strength distribution, the streamer formed near the cathode has a spherical shape [27,29,30]. The critical electric field strength $E_{cr}$ required for the transition of a significant part of electrons to the runaway mode can be determined from the expression $\alpha(E_{cr}, p)d = 1$ ($\alpha$—the first Townsend coefficient; $p$—gas pressure; $d$—gap width) [35]. Since this criterion is valid for a uniform electric field and near-zero initial electron energy $\varepsilon_0$, an increase in $\varepsilon_0$ leads to a decrease in $E_{cr}$ [36,37].

An example of a picture of the development of the streamer discharge in the gap with the sharply non-uniform electric field strength distribution filled with atmospheric-pressure air is shown in Figure 3. These data were obtained with the same generator that was used in this study. The gap breakdown in that case occurred at the voltage amplitude close to

that in the idle mode (Figure 3b, dotted waveform). For $d$ = 8.5 mm the energy deposited to the discharge plasma within 15 ns was ≈10 mJ.

At the initial stage, the streamer had a spherical shape. This is confirmed by the shape of the plasma glow region in frame C1. Then, for some time, this spherical streamer continued to increase in dimensions until the electric field strength at its boundaries began to noticeably decrease. At this moment, the streamer propagated to the flat electrode mainly along the longitudinal axis of the discharge (frame C2). Frames C3 and C4 demonstrate an already formed diffuse discharge. It was seen that its transverse dimensions were ~2 times smaller than those of the streamer. The diffuse form of the discharge persisted throughout the voltage pulse duration, and there were no bright spots on the flat anode. A bright spot appeared at the cathode tip when the discharge current reached 10 A and more, see, e.g., [32]. An increase in $d$ from 6 (Figure 2b) to 8.5 mm (Figure 3b) led to an increase in the breakdown voltage and a decrease in the discharge current.

Thus, the presented results show that for certain excitation parameters, the gap width, and the electrode design, bright thin current channels are not observed in the diffuse discharge, and bright spots are not formed on the anode. In addition, measurements performed with a current collector indicate the generation of runaway electrons under these conditions, see the waveforms in [38]. A decrease in the transverse dimensions of the cathode and the radius of curvature of its tip contribute to an increase in transverse dimensions of the spherical streamer and better discharge uniformity. However, if the dimensions are too small, the tip quickly melts and the radius of curvature of the cathode increases, see, e.g., [33]. Therefore, the conical stainless-steel cathode with the apex angle of ≈50° and the radius of curvature of ≈0.2 mm was used in experiments to study damage to a copper anode with and without carbon particles deposited on it. It should be noted that a decrease in the voltage pulse duration and the absence of voltage pulses reflected from the generator led to an improvement in the discharge uniformity.

*3.2. The Effect of Diffuse and Spark Discharges on the Anode during the Breakdown of Atmospheric Air in an Inhomogeneous Electric Field*

In the experiments, the uncoated and carbon-coated (20 μm thickness layer) copper anodes were exposed to various discharge morphologies. At $d$ = 8–10 mm the discharge was diffuse and had a conical shape near the pointed electrode (cathode), smoothly turning into a cylindrical one as it approached the anode (Figure 4a–c). A relatively small bright spot is seen on the pointed cathode, while the absence of any spots is observed on the anode. The images in Figure 4a–c correlate with those from the ICCD camera in Figure 3a (frame C4). At these $d$, plasma action did not lead to surface erosion of the uncoated copper and more sensitive aluminum anodes, as well as the carbon layer (Figure 4d). In this case, there was a homogeneous effect of the discharge plasma on the surface of the flat grounded electrode.

All other things being equal, a decrease in the gap width leads to changes in the discharge morphology and the results of the discharge plasma action on the anode (Figure 5). With increasing $I$ and, as a sequence, $P$, a bright glow near the anode surface arose. This caused damage to part of the carbon layer on the anode surface in contact with the plasma. Characteristic waveforms of $U$, $I$, and $P$ for this mode are presented in Figure 2b. Damage to the carbon layer was characterized by many craters, the diameters of which varied in the range of 10–50 μm (Figure 5c). The single largest crater, denoted as $C2$, was about 120 μm in diameter. The images show that the bottom of the craters was mostly cleared of carbon particles and the surface of the copper anode was clearly visible. Spark leaders near the anode and spark channels against the background of the diffuse discharge were not observed under these conditions. However, in the presence of a carbon layer on the anode, a bright glow appeared near its surface (Figure 5a). It can be explained by the combustion of carbon particles in the presence of oxygen. No damages were found on the surface of the copper anode. Additionally, with the uncoated anode, there was no bright glow near its surface, and the discharge morphology was the same as for $d$ = 8 mm in Figure 4b.

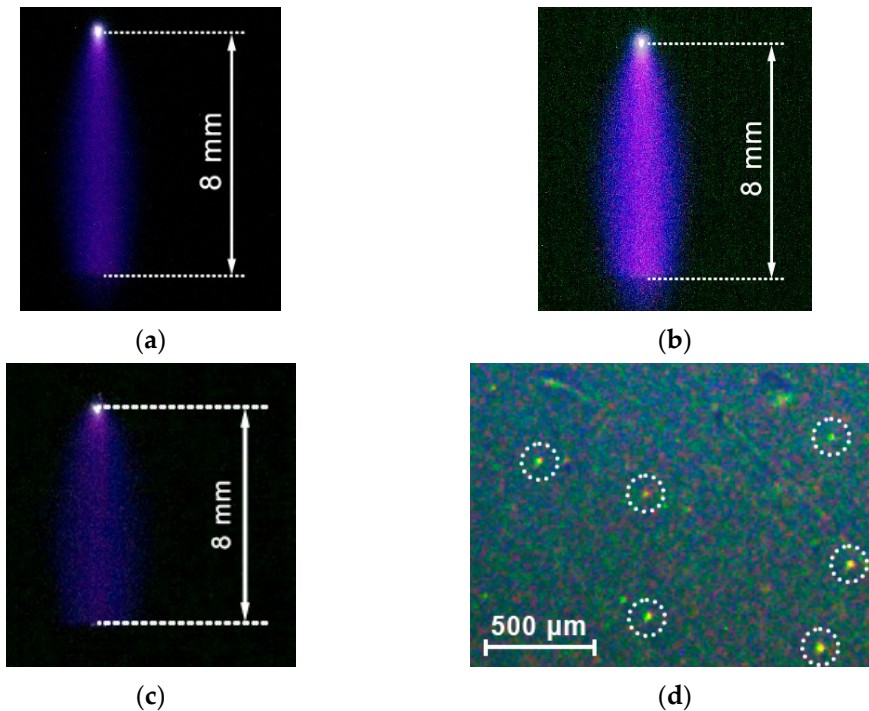

(a)

(b)

(c)

(d)

**Figure 4.** Integral images of the discharge plasma glow in air at *p* = 1 atm and *d* = 8 mm (**a**–**c**). Uncoated copper anode at the standard (**a**) and increased contrast (**b**). Carbon-coated (20 μm thickness layer) anode at the standard contrast (**c**). Imprint on the carbon layer (**d**). Dotted circles (see the imprint) denote dust particles. Single-pulse mode.

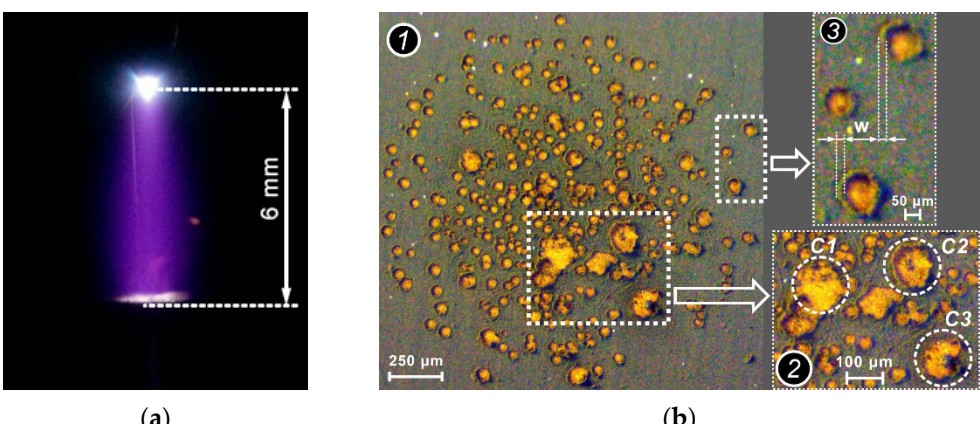

(a)

(b)

**Figure 5.** Integral image of the discharge plasma glow in atmospheric-pressure air at *d* = 6 mm captured at the standard contrast (**a**). Discharge plasma imprint on the flat electrode captured with additional lighting transversely to its surface (**b**). *1*—photograph of the entire surface; insets *2*, *3*—magnified images of areas selected indicated by the dotted frames. *C1*, *C2*, *C3* denote craters; w denotes the thickness of the layer around the crater formed as a result of damage. For this, an additional illumination was directed to the anode. The carbon layer thickness is ≈20 μm. Single-pulse mode.

We assume that a large number of craters is related to the electric field enhancement between the anode and the uncompensated space charge of positive ions remaining after electrons go to the anode. This electric field, directed oppositely to the external one, initiates the explosive emission from the anode surface covered with the carbon layer. With an increase in the electric field strength, this mechanism leads to the appearance of bright spots on metal anodes [11,20].

Further decrease in *d* provided the more rapid formation of spark leaders and channels, damaging both carbon-coated and uncoated anode surfaces. Figure 6 demonstrates integral images of the discharge plasma glow captured through the side window and imprints of the discharge plasma exposure on the uncoated and carbon-coated anode surfaces. In the case of the carbon-coated anode, the discharge, in about half of the implementations, continued to burn in the diffuse form, i.e., spark channels were not formed (Figure 6a). However, the brightness of the plasma glow near the anode increased. Damage to the carbon layer on the anode can be conditionally divided into several zones (they are shown in Figure 6d). In the center, the coating was destroyed and removed from a large area (see Figure 6b). Note that the damage in the central part did not have the shape of a crater both during the formation of the spark channel and during the appearance of a glow near the anode. In the second part of zone (*1*), surrounding the central damage and having the shape of a ring (Figure 6b), a large number of craters with a diameter of 10–50 μm appeared, similar to those shown in Figure 5c. This was followed by the second, also annular zone (*2*), in which the number of craters was small and the carbon layer remained largely intact. In addition, partial damage to the carbon layer was visible at the distance of 3–4 mm from center of discharge (zone (*3*)). They look like parts of a ring, which can be explained by the impact of a shock wave, which has a stronger effect at small gaps and the formation of a spark channel. Note that the photographs in Figure 6a,b correspond to the same pulse.

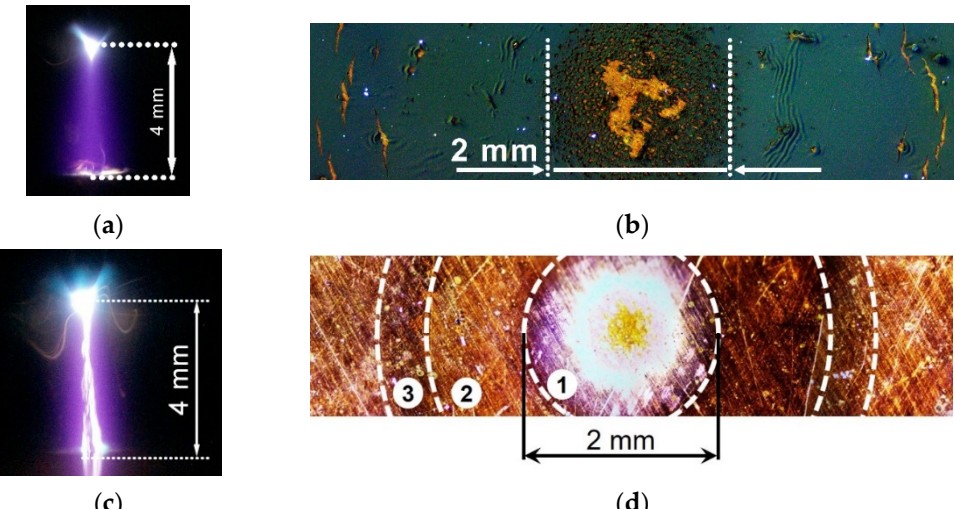

(**a**)                               (**b**)

(**c**)                               (**d**)

**Figure 6.** Integral images of the discharge plasma glow captured through the side window (**a**,**c**) and imprints of the discharge plasma action on the anode surface (**b**,**d**). Carbon-coated anode, single pulse (**a**,**b**). Uncoated anode (**c**,**d**). Image (**d**) was taken for ten pulses. *d* = 4 mm. *1*, *2*, *3*–ring exposure zones.

When exposed to the uncoated anode, traces of the impact of spark channels were visible in its central part. Here several differently colored zones can be specified. With distance from the central axis, the discharge current density decreased, which led to a change in the color of the anode surface as a result of the action of several pulses. The diameter of the light spot with dark edge (*1*) corresponds to the transverse size of the discharge zone occupied by the plasma. At the same time, the diameters of the plasma exposure zone on the carbon-coated and uncoated anodes in Figure 6b,d approximately coincided. In addition, with a decrease in the gap width, the number of metal particles that were removed from the cathode increased. At the same time, the intensity of their glow also increased. A decrease in *d* also led to a change in the color of the plasma glow at the tip of the high-voltage cathode made of stainless steel. In Figure 6c, a blue glow is observed at the cathode. A similar blue glow near stainless-steel electrodes was detected in [39] during the spark discharge formation. This glow color is due to the spectral transitions of atomic iron.

Cathode erosion for a large number of pulses (several hundred) at various *d* is shown in Figure 7a. This image was taken under additional illumination through a transparent conductive anode. The image of the discharge plasma glow captured in the cathode plane (Figure 7b) demonstrates a bright white cathode spot formed due to the explosive emission [33]. The spot is surrounded by a diffuse discharge plasma, against the background of which tracks of individual microparticles ejected from the cathode are distinguishable. Similar tracks were also observed in the photographs of the discharge obtained through the side windows (Figure 6c). The emission of the discharge plasma at the anode plane is uniform (Figure 7c). To minimize damage to the transparent conducting anode, no carbon layer was deposited on its surface, and the discharge was also ignited in short gaps. The photos in Figure 7 were captured with a SIGMA 50 mm 1:2.8 DG MACRO macrolens with a minimum depth of field.

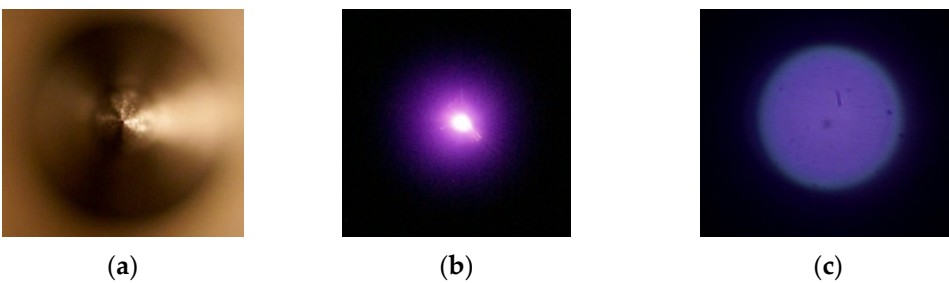

(**a**)  (**b**)  (**c**)

**Figure 7.** Images of the discharge gap captured through the transparent anode. Cathode after several hundred of discharges (**a**). Discharge plasma glow within single pulse in the cathode (**b**) and anode planes (**c**). *d* = 4 (**a**) and 6 (**b**,**c**) mm.

Spectral studies of the plasma glow showed that in the case of the diffuse discharge the distance between the electrodes for the setup in Figure 1 should be at *d* = 8–10 mm. Plasma emission spectra for *d* = 6 mm is shown in Figure 8.

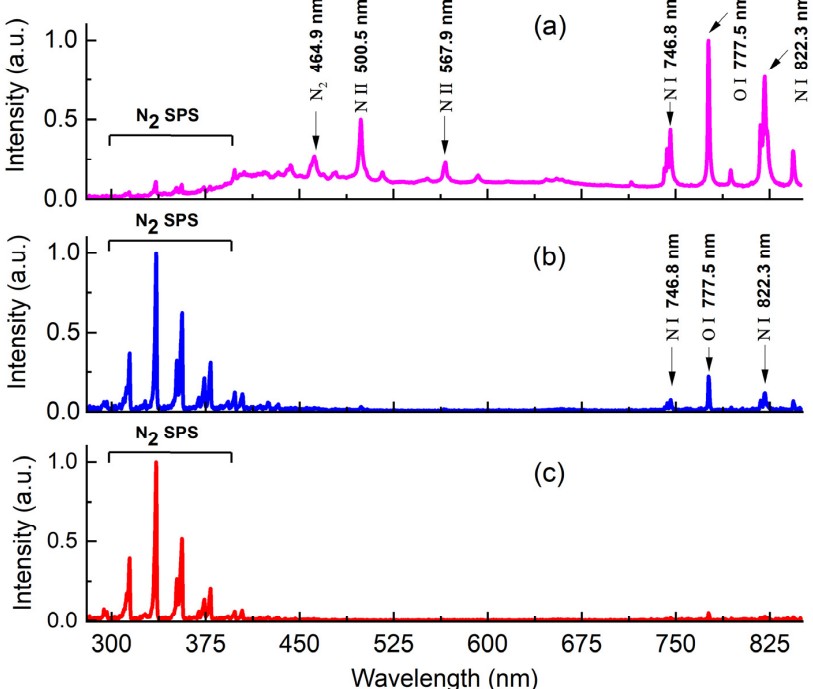

**Figure 8.** Discharge plasma emission spectra registered from the following regions in the 6-mm-width gap: near the pointed stainless-steel cathode (**a**), gap center (**b**), near the copper uncoated flat anode (**c**).

The spectra were recorded from the near-cathode, central, and near-anode regions of the gap. The pointed cathode was made of stainless steel. The anode was an uncoated copper plate. The anode was not covered with a carbon film, since it was necessary to make several tens of pulses to obtain one emission spectrum. At *d* = 6 mm, a spark leader similar to that shown in Figure 4a,b was observed at the cathode. The presence in the spectrum of a broadband continuum in the range of 300–900 nm (Figure 8a) is due to the emission of the leader formed near cathode. SPS dominated in the emission spectrum of the diffuse discharge plasma in the middle of the gap and in the near-anode region (Figure 8b,c). The most intense bands were those with wavelengths of 315.9, 337.1, 357.7, and 380.5 nm. When spark channels were formed in the gap with $d \leq 2$ mm, the bulk of the plasma radiation energy was concentrated in a wide band in the visible range of the spectrum. Under these conditions, a large number of craters appeared on the anode, as in [11–15].

## 4. Discussion and Conclusions

It is known [11–19,22,23] that the spark stage is formed at short gaps and high overvoltage on time scales $\leq 1$ ns. In this case, thin channels with a high electron concentration appear, first of all, near the cathode, and erosion traces of various shapes appear on the surface of electrodes. The emission spectra of the spark discharge plasma contain the recombination radiation of hot plasma, as well as lines and bands of vapors and metal particles (see Figure 8b). From the point of view of the practical application of diffuse discharges, it is necessary to ensure uniform treatment of a flat anode (target). Studies carried out by depositing a layer of carbon micro- and nanoparticles on the anode confirm the possibility of non-erosive (non-destructive) action of a diffuse discharge on the surface of a target-sample (anode). Under these conditions, a relatively "cold" plasma is formed, the electron temperature of which significantly exceeds the temperature of heavy particles. The emission spectrum of the diffuse discharge in air is shown in Figure 8a. The highest intensity was observed for the bands of SPS, which was observed under similar conditions in a great number of studies, see, e.g., [20,29]. To achieve such a treatment mode, it is necessary to form a diffuse discharge in an inhomogeneous electric field with a cathode having a small radius of curvature by applying short-duration high-voltage pulses to it. Under these conditions, the gap breakdown occurs as a result of the development of a wide streamer, which initially has a spherical shape and then stretches along the longitudinal axis of the discharge gap. Wide streamers have large transverse dimensions and their dynamics differ significantly from those of classical small-diameter cylindrical streamers observed in a uniform electric field (see, e.g., [34,40]). Nevertheless, in atmospheric pressure air, the electron density for both streamer types does not exceed $\sim 10^{14}$ cm$^{-3}$ [40,41]. However, a small-diameter streamer in a uniform electric field, due to subsequent ionization waves, passes into a spark channel, which has a local effect on the anode. The duration of the diffuse stage of the discharge during a breakdown in a non-uniform electric field increases. The cross-section of the plasma exposure region on the anode also increases and its uniformity is preserved. The main feature ensuring the formation of diffuse discharges in a non-uniform electric field is the generation of runaway electrons (and bremsstrahlung X-rays), which provide additional preionization of the discharge gap [32,38]. Note that, during the formation of wide streamers, the formation of a spark channel is preceded by the appearance of a bright s pot on the anode and the growth of a cathode-directed spark leader from it against the background of the diffuse discharge [20].

Let us describe again the phases of the diffuse discharge formation with the experimental setup shown in Figure 1. In a non-uniform electric field at high overvoltages, the discharge formation begins near a cathode with a small radius of curvature. The first electrons appear due to the field emission, the threshold of which can be additionally reduced due to the presence of microinhomogeneities and micro-and nanoparticles from various materials on the cathode surface. Primary electrons provide the formation of electron avalanches, the heads of which overlap at small distances from the pointed cathode due to the high electric field strength near its tip. When the critical number of electrons

in the avalanche is reached, a streamer arises. At this time point, some of the electrons go into the runaway mode and preionize neutral gas in the gap. At the initial stage, the streamer, as noted above, has a spherical shape, and then stretches along the longitudinal axis, taking a conical shape. An example of the development of a streamer discharge in air at atmospheric pressure is shown in Figure 3. The diameter (transverse dimensions) of the diffuse discharge at the anode was ~2 mm. Under conditions of this experiment, at $d$ = 8 mm or more, plasma exposure on the anode was soft—the carbon film was not damaged (Figure 4d). However, it should be noted that with an increase in the current density of the diffuse discharge and maintaining its uniformity, the electric field changes its direction as the streamer head approaches the anode. This occurs due to the departure of the main part of the electrons to the anode, while the ions, whose mobility is orders of magnitude lower, remain practically immobile and create an area of excess positive charge. Since the electric field strength between the anode surface and the positive ions cloud is directed in the opposite direction, the process of explosive emission begins [33], which leads to an increase in the current from microinhomogeneities on the anode. Accordingly, the carbon film is damaged (Figure 5b) and, as shown in this paper, ignites (Figure 5a). Estimated by the maximum discharge currents in Figures 2b and 3b showed that the average current density at $d$ = 6 mm increased by a factor of 1.5 compared with $d$ = 8.5 mm.

Further, the discharge mode depends on the gap width, as well as the amplitude and duration of a voltage pulse. With decreasing gap width and (or) increasing duration and amplitude of the voltage pulse, spark leaders begin to grow from one or both electrodes. After the gap is closed by a spark leader or two counter leaders, the discharge passes into the spark stage. Accordingly, to modify the anode, it is necessary to use a diffuse form of the discharge without bright spots on a flat electrode, respectively, without damaging it. However, it should be taken into account that damages can occur due to the appearance of an electric field directed towards the anode near its surface due to the closing of the gap by the streamer front. Such damages on the carbon film are visible at a distance from the anode center (Figure 6d).

To increase the plasma diameter at the anode when the streamer is closed, it is necessary to increase the voltage pulse amplitude and the interelectrode distance [41] or use cathodes consisting of a large number of points [21].

Thus, in this work, covering a flat anode with a thin carbon layer, it was shown that before the appearance of bright local anode spots and the subsequent appearance of a spark channel, the anode is affected by a diffuse discharge plasma formed by closing the gap with a wide streamer. In addition, it was found that with a decrease in the interelectrode distance, the deposition of a carbon layer on the anode improves the uniformity of the discharge, and an increase in the diffuse discharge current density leads to the ignition of the carbon particles.

It should be noted in conclusion that the modes of diffuse discharges that can be used to treat the surfaces of metals and dielectrics are described in [21,27,29,32,41–43] and others.

**Author Contributions:** Conceptualization, writing—original draft, V.T. and M.L.; experiments, M.L. and D.B.; analysis, M.S.; methodology, M.L., M.S., and D.B.; review & editing, D.S. All authors have read and agreed to the published version of the manuscript.

**Funding:** This research was supported by the Ministry of Science and Higher Education of the Russian Federation within Agreement no. 075-15-2021-1026 of 15 November 2021.

**Informed Consent Statement:** Informed consent was obtained from all subjects involved in the study.

**Data Availability Statement:** Data are contained within the paper.

**Conflicts of Interest:** The authors declare no conflict of interest. The funders had no role in the design of the study; in the collection, analyses, or interpretation of data; in the writing of the manuscript, or in the decision to publish the results.

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
