# Peer review of "Nano-and Microparticles of Carbon as a Tool for Determining the Uniformity of a Diffuse Discharge Exposure"

_surfaces, doi:10.3390/surfaces6010004_

Round 1

Reviewer 1 Report

This work presents very interesting and innovative results of using carbon microparticles to illustrate the uniformity of pulsed discharge.

This work can be published after some modification.

1. The resistance value for the shunt of reverse current should be given.

2. What is the particle size distribution of the carbon coating?

3. How adhesive is connection between the coated carbon layer and the anode?

4. Figure 8, spectral bands/lines should be labeled

5. Is the color bar for images in Figure 3 the same?

6. The following literature, related to pulsed discharge uniformity, may be interesting to authors. Plasma Sources Sci. Technol. 31 (2022) 114002. J. Phys. D: Appl. Phys. 55 (2022) 235204. 

Author Response

Comments and Suggestions for Authors

This work presents very interesting and innovative results of using carbon microparticles to illustrate the uniformity of pulsed discharge.

This work can be published after some modification.

  1. The resistance value for the shunt of reverse current should be given.

Reply:

The resistance of the current shunt was 0.014 Ω.

  1. What is the particle size distribution of the carbon coating?

Reply:

The minimum size of carbon particles, which had a round shape when illuminated from the side of a transparent plate, was ≈700 nm at the beginning of the deposition procedure. It was also possible to distinguish particles with a diameter of about three times smaller. As the deposition time increased, the diameter of the round particles increased to ≈2 µm and they became opaque. A further increase in the deposition time led to a change in the shape of the particles and an increase in their size. Then the entire surface was covered with an opaque carbon film.

  1. How adhesive is connection between the coated carbon layer and the anode?

Reply:

Since the carbon film had low strength, it was not possible to determine its adhesion under these conditions. When the adhesive tape was glued to the carbon layer and then peeled off, the carbon layer remained both on the adhesive tape and on the anode surface.

  1. Figure 8, spectral bands/lines should be labeled.

Reply:

Figure 8 has been improved. Emission spectra of the discharge plasma recorded from the near-cathode, central, and near-anode regions of the 6-mm-width gap are demonstrated. The most intense spectral lines and bands are denoted. The caption has been revised.

  1. Is the color bar for images in Figure 3 the same?

Reply:

The ICCD images in Figure 3 were colored. Color bars were added to them.

  1. The following literature, related to pulsed discharge uniformity, may be interesting to authors. Plasma Sources Sci. Technol. 31 (2022) 114002. J. Phys. D: Appl. Phys. 55 (2022) 235204.

Reply:

The first of the specified references has been added to the list of sources. Instead of the second one, another reference affiliated with same scientific team has been added – to a newer publication related to the pulsed discharge uniformity.

The authors are grateful to the Referee for the comments and ask to support the publication of this study.

From authors:

Victor F. Tarasenko

Reviewer 2 Report

Journal: Surfaces,

MDPI Surfaces Editorial Office,

Mr. Grayson Li

Dear Mr. Grayson Li,

I hereby send comments on the manuscript, “Nano- and microparticles of carbon as a tool for determining the uniformity of a diffuse discharge exposure”, (Manuscript ID: surfaces-2188914). Authors: M. Lomaev, V. Tarasenko, M. Shulepov, D. Beloplotov, D. Sorokin.

 In this manuscript (M) authors considered the diffuse discharge plasma, in air at atmospheric pressure. This type of discharge is interesting from the reason of a surface treatment of various materials.

In context of the M the damages of the electrode-anode was especially observed. Thus, exposure uniformity was investigated by using nano- and micro-sized carbon particles deposited on a flat copper anode (a carbon layer). Among other, it was established that at given gap width (8–10 mm) the discharge plasma imprint on the carbon layer deposited on a copper anode has no traces of local erosion.

 In the main, the studies are interesting especially because the data in this area which are insufficiently known in literature. The results can be of interest for scientists/engineers which are involved in this area, thus:

 My general conclusion is, That manuscript can be accepted for publication.

Author Response

Comments and Suggestions for Authors

I hereby send comments on the manuscript, “Nano- and microparticles of carbon as a tool for determining the uniformity of a diffuse discharge exposure”, (Manuscript ID: surfaces-2188914). Authors: M. Lomaev, V. Tarasenko, M. Shulepov, D. Beloplotov, D. Sorokin.

 In this manuscript (M) authors considered the diffuse discharge plasma, in air at atmospheric pressure. This type of discharge is interesting from the reason of a surface treatment of various materials.

In context of the M the damages of the electrode-anode was especially observed. Thus, exposure uniformity was investigated by using nano- and micro-sized carbon particles deposited on a flat copper anode (a carbon layer). Among other, it was established that at given gap width (8–10 mm) the discharge plasma imprint on the carbon layer deposited on a copper anode has no traces of local erosion.

 In the main, the studies are interesting especially because the data in this area which are insufficiently known in literature. The results can be of interest for scientists/engineers which are involved in this area, thus:

 My general conclusion is, That manuscript can be accepted for publication.

The authors are grateful to the referee for supporting the publication of this study.

From authors:

Victor F. Tarasenko
